# Surgical care in district hospitals in sub-Saharan Africa: a scoping review

Zineb Bentounsi ,[1] Sharaf Sheik-Ali,[2] Grace Drury,[1] Chris Lavy [1]

► Prepublication history and additional materials for this paper is available online. To view these files, please visit the journal online (http://dx.doi. org/10.1136/bmjopen-2020-042862).

## ABSTRACT

**Objective** To provide a general overview of the reported current surgical capacity and delivery in order to advance current knowledge and suggest targets for further development and research within the region of sub-Saharan Africa.

**Design** Scoping review.

**Setting** District hospitals in sub-Saharan Africa.

**Data sources** PubMed and Ovid EMBASE from January 2000 to December 2019.

**Study selection** Studies were included if they contained information about types of surgical procedures performed, number of operations per year, types of anaesthesia delivered, cadres of surgical/anaesthesia providers and/or patients' outcomes.

**Results** The 52 articles included in analysis provided information about 16 countries. District hospitals were a group of diverse institutions ranging from 21 to 371 beds. The three most frequently reported procedures were caesarean section, laparotomy and hernia repair, but a wide range of orthopaedics, plastic surgery and neurosurgery procedures were also mentioned. The number of operations performed per year per district hospital ranged from 239 to 5233. The most mentioned anaesthesia providers were non-physician clinicians trained in anaesthesia. They deliver mainly general and spinal anaesthesia. Depending on countries, articles referred to different surgical care providers: specialist surgeons, medical officers and non-physician clinicians. 15 articles reported perioperative complications among which surgical site infection was the most frequent. Fifteen articles reported perioperative deaths of which the leading causes were sepsis, haemorrhage and anaesthesia complications.

**Conclusion** District hospitals play a significant role in sub-Saharan Africa, providing both emergency and elective surgeries. Most procedures are done under general or spinal anaesthesia, often administered by non-physician clinicians. Depending on countries, surgical care may be provided by medical officers, specialist surgeons and/or non-physician clinicians. Research on safety, quality and volume of surgical and anaesthesia care in this setting is scarce, and more attention to these questions is required.

[1]Nuffield Department of Orthopaedics, Rheumatology and Musculoskeletal Sciences, University of Oxford, Oxford, UK
[2]Oxford University Hospitals NHS Foundation Trust, Oxford, UK

**Correspondence to**
Dr Zineb Bentounsi;
zbent258@gmail.com

## Strengths and limitations of this study

► This is the first review (of which we are aware) to systematically assess the literature for the scope, volume and quality of surgery in district hospitals in sub-Saharan Africa.

► Article selection and data extraction were conducted independently by two authors.

► Our search was conducted in English, French and Portuguese, and work in other languages may have been omitted.

► This study is prone to publication bias as it includes only data from published literature, and findings that were never published will have been omitted.

► There was a lot of heterogeneity in how articles reported surgical volume and quality.

healthcare facilities that provide in-patient surgery and anaesthesia.[2] These hospitals mainly serve rural populations and face challenges in terms of workforce, infrastructure and other essential resources. In 2010, it was estimated that there was only 1 surgeon for 25 million people in rural sub-Saharan Africa[3] and although training programmes in the region are increasing the number of surgeons, the main burden in several countries remains on non-physician clinicians to manage surgical cases.[4]

District hospitals offer a unique opportunity to improve access to surgical care in sub-Saharan Africa as they are the first point of access to surgical care for patients and relieve referral hospitals from undue burden, allowing them to focus on complicated cases. Delivering timely surgical care in district hospitals has been shown to be highly cost effective,[5] which should provide encouragement for governments and global health programmes to invest in them. The Lancet Commission on Global Surgery,[6] which defines scalable solutions for the provision of quality surgical and anaesthesia care for all, reinforces this emphasis on district hospitals and echoes other authors' calls to consider district hospitals as a central component in strengthening surgical care in sub-Saharan Africa.[7 8]

## INTRODUCTION

The burden of surgical disease in sub-Saharan Africa is estimated at 38 disability adjusted life years per 1000 population. District hospitals bear a large burden of this surgical care delivery.[1] According to The WHO, district hospitals are defined as the first level of

We considered a systematic review of district hospital surgery; however, there were a limited number of existing studies in this field and a wide range of methodology used. Therefore, we chose to change our review methodology and undertake a scoping review of surgical care delivery in district hospitals in sub-Saharan Africa. Our aim is to provide a general overview of the reported current surgical capacity and delivery in order to advance current knowledge and suggest targets for further development and research within the region of sub-Saharan Africa.

## METHODS

This scoping review follows the five-step methodology designed by Arksey and O'Malley[9] that offers a rigorous framework for reporting findings. We have also considered the recommendations of Levac et al[10] that supplement this framework.

### Identifying the research question

To guide our search of the literature, we defined the following questions:

#### Primary outcome

In district hospitals in sub-Saharan Africa, what types of surgical procedures are provided?

#### Secondary outcomes

In district hospitals in sub-Saharan Africa:
- ► How many surgical procedures are performed per year?
- ► Are there any measures of quality, safety or outcomes of surgery?
- ► Who provides anaesthesia and/or surgery?
- ► What type of anaesthesia is delivered?

### Identifying relevant studies

We searched PubMed and Ovid EMBASE on 18 December 2019 to retrieve literature about surgery in district hospitals in sub-Saharan Africa. The search strategy combined the exploded thesaurus terms 'surgery', 'surgical procedures, operative' and 'specialties, surgical' with the exploded thesaurus terms 'hospitals, district' and 'Africa south of the Sahara'. We searched references in English, French and Portuguese. We limited the search to references published after 1 January 2000 in order to provide a recent overview and, therefore, studies published before 2000 were not relevant for this purpose. We also searched Google Scholar and the Cochrane Library with the same search terms, but they did not contribute any new relevant references.

### Study selection

The most important inclusion criterion was the setting: a district-level hospital in sub-Saharan Africa. We included articles that provided information about either type of surgical procedures performed, number of operations per year, types of anaesthesia delivered, cadres of surgical/anaesthesia providers and/or patient outcomes.

We excluded articles if the setting was unclear or was not a district hospital and if they were strictly about costs or burden of disease. We excluded articles if the full text could not be retrieved. We included all surgical procedures except those typically done in outpatient departments, such as dental and ophthalmic procedures.

ZB and GD independently screened the abstracts using the web application Rayyan,[11] resolving any conflicts by dialogue between ZB, GD and CL. The process of selection followed the Preferred Reporting of Items for Systematic Reviews and Meta-Analyses (PRISMA) Statement.

### Charting the data

Z.B. developed the data charting form after consultation with all other authors. The form was piloted and then updated iteratively during the extraction process (see online supplemental file 1). The form contained 20 fields of data entry covering information about articles (eg, name of first author, year of publication, research method) and information about district hospitals, surgical and anaesthesia providers, surgical procedures as well as other relevant information. ZB and SS-A extracted the data independently.

### Collating, summarising and reporting the results

SS-A produced descriptive numerical summaries of surgical volume, types of surgical procedures and types of anaesthesia using Excel. ZB performed a thematic analysis in NVivo[12] with surgical providers, quality of care and surgical outcomes as themes. We reported results to answer the primary and secondary outcomes of the review. All authors discussed and reported implications for research and policy.

## RESULTS

In this section, we first present characteristics of the included articles and an analysis of the district hospitals. Thereafter, we present our findings on the scope and volume of surgery, types of anaesthesia, surgical and anaesthesia providers and finally safety and quality of care.

### Study selection

We identified 302 articles and after removing duplicates and 267 articles remained. We screened the titles and abstracts, and, using our criteria noted above, excluded 167 articles. Then, we assessed full texts of the remaining 100 articles and excluded 48. Therefore, we included 52 articles for analysis. A PRISMA diagram is presented in figure 1.

### Study characteristics

Thirty three per cent of articles (17 articles) were cross-sectional surveys, 17% were prospective studies (nine articles) and 11.5% (six articles) were based on retrospective review of hospital records (table 1). Ninety-three per cent of articles (48 articles) described the surgical care provided by staff based on the hospital, and the rest (four

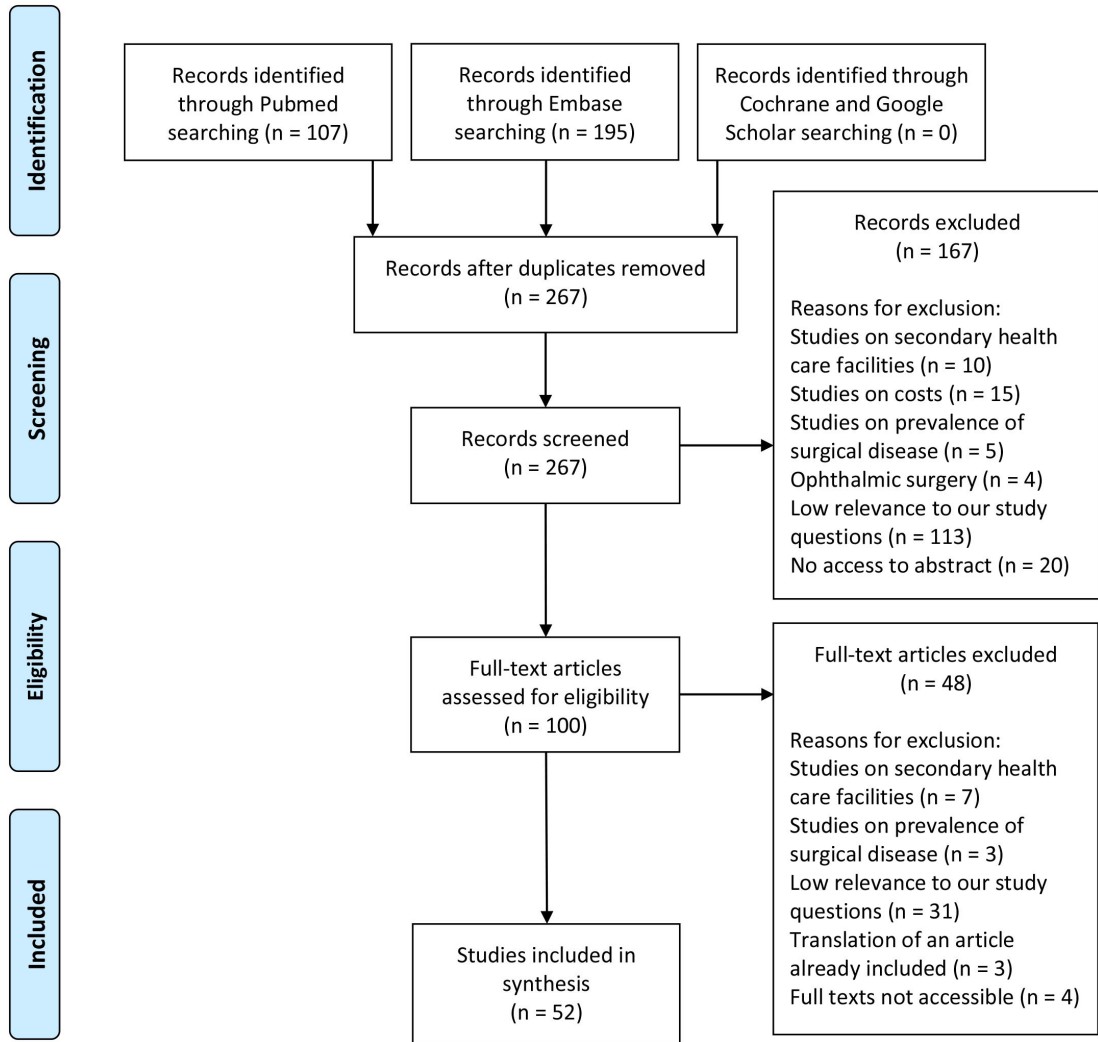

**Figure 1** PRISMA diagram illustrating the selection process. PRISMA, Preferred Reporting of Items for Systematic Reviews and Meta-Analyses.

articles) reported external support from nongovernmental organisations (NGO) or visiting surgical teams[13–17]

Of the 52 articles included for analysis, covering 16 countries, 3 articles presented data from multiple countries[18–20] and 2 gave a regional perspective.[21 22] Ghana, Malawi and Rwanda were the three countries most represented in the articles selected, partly because of research projects in collaboration with researchers from high-income countries (table 2).

### Characteristics of district hospitals

District hospitals described in this review included a variety of institutions (table 3). In our review, the two smallest district hospitals had 21 beds, reported in Liberia and Ghana by Sherman *et al*[23] and Choo *et al*,[24] respectively. The biggest district hospital had 371 beds, reported by Fehr *et al* in Tanzania.[25] District hospitals served rural or urban populations, with some serving both groups (for example, if they were situated on the edge of towns). Although most were owned by the government, mission-linked district hospitals also featured in several papers. On some occasions, hospitals received support from visiting

surgeons or NGOs, for example, in Rwanda, government-owned district hospitals received support from partners in health.[15]

### Scope of surgical operations

The majority of articles mentioned obstetrics/gynaecology and general surgery operations (table 4). Indeed, the three most frequently reported procedures were caesarean section, laparotomy and hernia repair. However, a wide range of other surgical procedures were mentioned across the literature (online supplemental file 2). These were mainly in orthopaedics, plastic surgery or neurosurgery and may have involved visiting specialists (table 4). District hospitals reported in the literature handled both emergency procedures (such as caesarean sections, appendicectomies and treatment of burns) and elective procedures (such as hernia repairs) and some even more complex procedures such as direct vision internal urethrotomy[26] and hip replacement.[14] Despite the high prevalence of traumatic brain injury in sub-Saharan Africa[27] and the fact that cranial burr holes are

**Table 1** Summary of studies selected

| First author | Year of publication | Study setting | Study design | Objective |
|---|---|---|---|---|
| Abdullah et al[30] | 2011 | Ghana | Mixed-methods | Assess surgical and obstetrical capacity |
| Cheelo et al[59] | 2018 | Zambia | Cross-sectional survey | Assess surgical capacity |
| Choo et al[61] | 2013 | Ghana | Cross- sectional survey | Appraise quality improvement activities |
| Choo et al[24] | 2011 | Ghana | Qualitative methods | Explore the training of medical officers providing surgery |
| Compaoré et al[31] | 2014 | Burkina Faso | Cross-sectional survey | Assess the readiness to provide caesarean sections |
| Damien et al[32] | 2011 | Ghana | Review of hospital records | Evaluate surgical volume |
| De Brouwere et al[56] | 2009 | Senegal | Mixed-methods | Evaluate the task shifting policy |
| Dossche et al[14] | 2014 | Burkina Faso | Case series | Analyse technical problems and post-operative complications |
| Dresser et al[58] | 2017 | Uganda | Review of hospital records | Evaluate the management of surgical patients by non-physician emergency clinicians |
| Fehr et al[25] | 2006 | Tanzania | Prospective cohort study | Identify risk factors for surgical site infection |
| Fehr et al[65] | 2006 | Tanzania | Prospective cohort study | Evaluate peri-operative antimicrobial prophylaxis |
| Fenton et al[39] | 2003 | Malawi | Prospective cohort study | Identify risk factors associated with maternal death after caesarean section |
| Gajewski et al[33] | 2017 | Malawi | Baseline and follow-up patients' surveys | Compare surgical outcomes between central and district hospitals |
| Gajewski et al[18] | 2019 | Malawi, Tanzania, Zambia | Mixed-methods | Investigate anaesthesia capacity |
| Gajewski et al[34] | 2018 | Malawi | Qualitative methods | Explore providers' perspectives on obstacles to surgery |
| Gajewski et al[35] | 2019 | Malawi | Randomised control trial | Evaluate surgical training programme for clinical officers |
| Galukande et al[19] | 2010 | Uganda, Tanzania, Mozambique | Review of hospital records | Assess the scope of surgery provided |
| Grimes et al[21] | 2012 | Sub-Saharan Africa | Systematic review | Identify the met and unmet needs of surgical disease |
| Gyedu et al[36] | 2018 | Ghana | Cross-sectional survey | Evaluate the operation rate in Ghana |
| Hall et al[37] | 2015 | Ghana | Review of hospital records | Evaluate the impact of a newly placed obstetrician on health outcomes |
| Harfouche et al[38] | 2015 | Malawi | Prospective cohort study | Assess the quality of emergency caesarean sections |
| Henry et al[22] | 2015 | Sub-Saharan Africa | Consensus meeting +interviews | Identify a package of essential surgical operations |
| Henry et al[55] | 2015 | Malawi | Cross-sectional survey | Assess surgical capacity |
| Kenfack et al[66] | 2012 | Cameroon | Case series | Describe the management of ectopic pregnancy |
| Koigi-Kamau et al[62] | 2005 | Kenya | Prospective cohort study | Describe the incidence of post-caesarean wound infection |
| Lavy et al[67] | 2007 | Malawi | Review of hospital records | Estimate surgical volume |
| LeBrun et al[20] | 2014 | Ethiopia, Liberia, Rwanda, Uganda | Cross- sectional survey | Assess surgical capacity |
| Lofgren et al[40] | 2015 | Uganda | Cross-sectional survey | Investigate indications, interventions and peri-operative mortality |
| Lonnée Lonn[66] | 2018 | Zimbabwe | Cross-sectional survey | Describe anaesthesia care for caesarean sections |
| Luo et al[45] | 2018 | Ghana | Qualitative methods | Explore the impact of a newly placed obstetrician in a district hospital |

**Table 1** Continued

| First author | Year of publication | Study setting | Study design | Objective |
|---|---|---|---|---|
| McCord[46] | 2012 | Ghana | Commentary | Describe surgical care in district hospitals in Ghana |
| McCord et al[57] | 2009 | Tanzania | Review of hospital records | Evaluate the quality of emergency obstetric surgery done by assistant medical officers |
| Mehtsun et al[47] | 2012 | Ghana | Observational study | Assess surgical workload of medical officers |
| Mpirimbanyi et al[17] | 2017 | Rwanda | Cross-sectional survey | Describe the spectrum, management and patient outcomes of emergency surgery |
| Muhirwa et al[15] | 2016 | Rwanda | Cohort study | Describe the presentation, management and outcomes of surgical patients |
| Ngaroua et al[26] | 2017 | Cameroon | Case series | Describe the management of urethral stenosis |
| Nkurunziza et al[16] | 2019 | Rwanda | Cross-sectional survey | Estimate the prevalence of surgical site infection after caesarean section |
| Nordberg et al[49] | 2002 | Kenya | Cross-sectional survey | Estimate the surgical volume |
| Notrica et al[50] | 2011 | Rwanda | Cross-sectional survey | Assess surgical and anaesthesia infrastructure |
| Ottaway et al[41] | 2018 | Namibia | Prospective Observational Study | Determine the prevalence of anaesthetic adverse events |
| Ouédraogo et al[42] | 2015 | Burkina Faso | Prospective Observational Study | Describe task shifting in obstetric care |
| Ouédraogo et al[43] | 2015 | Burkina Faso | Case series | Describe prognosis of patients undergoing caesarean sections |
| Petroze et al[51] | 2012 | Rwanda | Cross-sectional survey | Evaluate the ratio of caesarean sections to total procedures as a marker of trauma capacity |
| Petroze et al[52] | 2012 | Rwanda | Mixed-methods | Assess emergency and essential surgical capacity |
| Ramos et al[13] | 2013 | Ethiopia | Case series | Describe thyroid surgery cases |
| Rutgers and VanEygen[53] | 2008 | Zimbabwe | Cross-sectional survey | Describe the mortality associated with caesarean sections |
| Sani et al[54] | 2009 | Niger | Cross-sectional survey | Assess surgical care provided by general practitioners |
| Sherman et al[23] | 2011 | Liberia | Cross-sectional survey | Assess emergency and essential surgical care |
| Smiley et al[48] | 2019 | Ghana | Surveys+process mapping | Evaluate patients and staff perception of quality of care + analyse peri-operative process |
| Stewart et al[63] | 2015 | Ghana | Cross-sectional survey | Assess trauma capacity |
| Van Amelsfoort et al[29] | 2010 | Malawi | Mixed-methods | Describe the training and experiences of clinical officers |
| Van den Akker et al[64] | 2009 | Malawi | Mixed-methods | Evaluate the impact of audits in reducing the incidence of uterine ruptures |

included in the WHO book 'Surgical Care at the District Hospital',[28] only three articles mentioned burr holes.

### Surgical volume

Only 13 articles provided enough data to be able to estimate the number of operations per year per district hospital performed. In some cases, there was not enough information to discriminate between major and minor operations, therefore, the choice was made to present all operations combined. Using a simple formula, the number of operations, period of years and number of district hospitals, an average of the number of operations per district hospital per year was calculated (table 5). Galukande et al[19] and LeBrun et al[20]

contributed several reports on different district hospitals while van Amelsfoort et al[29] contributed two reports on the same district hospital during two different time periods.

The surgical volume of district hospitals varied from circa 200 to circa 5000 operations per year, with the majority of hospitals performing between 1001 and 5000 operations per year (tables 5 and 6). Clear limitations to the use of these data are the sample size and selection bias. Of the 13 studies, only 6 analysed more than five district hospitals in a given area. These data, therefore, cannot solely be used as a general indicator for the productivity of district hospitals in the countries represented.

| Table 2 | Distribution of articles per country |
|---|---|
| **Country** | **Number of articles** |
| Burkina Faso | 3 |
| Cameroon | 2 |
| Ethiopia | 2 |
| Ghana | 11 |
| Kenya | 2 |
| Liberia | 2 |
| Malawi | 10 |
| Mozambique | 1 |
| Namibia | 1 |
| Niger | 1 |
| Rwanda | 7 |
| Senegal | 1 |
| Sub Saharan Africa (regional perspective) | 2 |
| Tanzania | 5 |
| Uganda | 4 |
| Zambia | 2 |
| Zimbabwe | 2 |

### Types of anaesthesia

Out of 52 articles, 16 contained information about the type of anaesthesia used in the district hospitals studied. General anaesthesia was mentioned by 15 articles[13 15 17–19 21 30–38] and spinal anaesthesia by 10 articles.[14 17 23 24 39–44] Local anaesthesia and regional blocks were mentioned by four articles.[15 17 23 40]

### Surgical and anaesthesia providers

In each country, different cadres of healthcare professionals are responsible for providing surgical care in district hospitals (table 7). In Ethiopia, Ramos *et al* reported the presence of a specialist surgeon.[13] In Ghana,[24 30 36 37 45–48] Kenya,[49] Rwanda[15–17 50–52] and Zimbabwe[53] medical officers worked alongside specialist surgeons or obstetrician/gynaecologists to provide surgical care. In Namibia[41] and Niger,[54] the presence of medical officers as sole surgical providers was reported. Non-physician clinicians provided surgical care in Burkina Faso,[14 31 42] Liberia,[23]

| Table 3 | Distribution of articles per size of district hospitals |
|---|---|
| **Size of district hospitals (in number of beds)** | **Number of articles** |
| <20 | 0 |
| 20–50 | 2 |
| 51–100 | 8 |
| 101–300 | 9 |
| >300 | 2 |
| Not specified | 33 |

| Table 4 | Distribution of articles per surgical operations |
|---|---|
| **Surgical procedures** | **Number of articles** |
| General surgery | |
| Hernia repair | 26 |
| Laparotomy | 21 |
| Appendicectomy | 10 |
| Abscess | 7 |
| Hydrocele repair | 7 |
| Thyroid surgery | 5 |
| Colectomy | 4 |
| Chest tube insertion | 4 |
| Prostatectomy | 4 |
| Cricothyroidotomy | 3 |
| Colostomy | 2 |
| Lipoma | 2 |
| Tracheostomy | 2 |
| Fibroma | 1 |
| Urethrotomy | 1 |
| Vascular repair | 1 |
| Bladder stone surgery | 1 |
| Obstetrics and gynaecology | |
| Caesarean section | 30 |
| Salpingectomy (ectopic pregnancy) | 10 |
| Hysterectomy | 9 |
| Evacuation of retained products of conception | 7 |
| Tubal ligation | 5 |
| Fistula repair | 3 |
| Cervical tear repair | 2 |
| Breast surgery | 1 |
| Orthopaedics | |
| Amputation | 10 |
| Open fracture repair | 8 |
| Closed fracture repair | 5 |
| Arthrotomy | 3 |
| Osteotomy | 2 |
| Total hip replacement | 1 |
| Bipolar hemiarthroplasty | 1 |
| Neurosurgery | |
| Major neurosurgery | 1 |
| Burr holes | 3 |
| Plastics | |
| Skin grafting | 7 |
| Debridement | 5 |
| Burn | 4 |
| Cleft lip/palate surgery | 3 |
| Contracture release | 2 |
| Escharotomy | 1 |

**Table 5** Surgical volume (all reported cases) in number of operations per year per district hospital

| First author | Study setting | Total number of operations | Years | Number of district hospitals | Estimated number of operations per district hospital per year (to the nearest unit) |
|---|---|---|---|---|---|
| Abdullah*[30] | Ghana | – | 1 | 10 | 774 |
| Damien[32] | Ghana | 1391 | 5 | 1 | 278 |
| Galukande | Tanzania | 980 | 1 | 1 | 980 |
| Galukande | Tanzania | 2045 | 1 | 1 | 2045 |
| Galukande | Mozambique | 601 | 1 | 1 | 601 |
| Galukande | Mozambique | 256 | 1 | 1 | 256 |
| Galukande | Uganda | 1484 | 1 | 1 | 1484 |
| Galukande | Uganda | 248 | 1 | 1 | 248 |
| Galukande | Uganda | 239 | 12 | 1 | 239 |
| Galukande | Uganda | 1835 | 1 | 1 | 1835 |
| Gyedu | Ghana | 143750 | 1 | 48 | 2995 |
| Lofgren † | Uganda | – | – | 2 | 5018 |
| Lavy[67] | Malawi | 25053 | 1 | 21 | 1193 |
| LeBrun‡ | Uganda | – | 1 | 1 | 1296 |
| LeBrun‡ | Rwanda | – | 1 | 1 | 2052 |
| LeBrun‡ | Liberia | – | 1 | 1 | 696 |
| LeBrun‡ | Ethiopia | – | 1 | 1 | 538 |
| Nordberg* | Kenya | – | 1 | 5 | 5233 |
| Notrica‡ | Rwanda | – | – | – | 2052 |
| Ottaway | Namibia | 737 | 0.58 | 4 | 316 |
| Petroze | Rwanda | 71432 | 1 | 41 | 1742 |
| Sani | Niger | 544 | 1 | 3 | 181 |
| Van Amelsfoort | Malawi | 19644 | 1 | 17 | 1156 |
| Van Amelsfoort | Malawi | 18524 | 1 | 17 | 1090 |

*The article reported numbers for both major and minor operations and we calculated the total number of procedures.
†This article reported that the number of major operations was estimated over a 4 month period while the number of minor operations was estimated over a 3 month period.
‡Reported in the article as average number per year per district hospital.

Malawi,[33–35 39 55] Senegal,[56] Tanzania,[18 57] Uganda[40 58] and Zambia.[59]

Similarly, different cadres performed anaesthesia in different countries.

Articles reported the presence of a physician anaesthetist in district hospitals in Burkina Faso[14] and Ghana.[48] Medical officers providing anaesthesia on their own were reported in Namibia.[41] Otherwise, the most mentioned

anaesthesia providers are non-physician clinicians trained in anaesthesia (table 8). Their presence is reported in Burkina Faso,[31] Ethiopia,[13] Ghana,[24] Liberia,[23] Malawi,[20 28]

**Table 6** Number of publications reporting the number of operations per year per district hospital

| Number of operations on average per district hospital over 1 year | Number of publications |
|---|---|
| <100 | 0 |
| 100–500 | 6 |
| 501–1000 | 5 |
| 1001–5000 | 11 |
| >5000 | 2 |

**Table 7** Number of publications reporting surgical providers

| Surgical providers | Number of publications |
|---|---|
| Specialist surgeon/obstetrician | 5 |
| Specialist surgeon/obstetrician & medical officers | 9 |
| Medical officers | 8 |
| Medical officers and non-physician clinicians | 4 |
| Specialist surgeon/obstetrician and non-physician clinicians | 1 |
| Non-physician clinicians | 5 |
| Specialists and medical officers and non-physician clinicians | 8 |

**Table 8** Number of publications reporting anaesthesia providers

| Anaesthesia providers | Number of publications |
|---|---|
| Physician anaesthetists | 2 |
| Medical officers | 1 |
| Medical officer and non-physician clinicians | 1 |
| Non-physician clinicians (trained) | 10 |
| Non-physician clinicians (trained and untrained) | 6 |

Rwanda,[51 56] Senegal,[56] Tanzania,[18] Uganda,[40] Zambia[59] and Zimbabwe.[57 60] There are also untrained non-physician clinicians providing anaesthesia in Liberia,[23] Malawi,[20 28] Rwanda,[51 56] Zimbabwe[57 60] alongside trained personnel.

## Quality of care and surgical outcomes
### Quality of care

Of all articles in our review, only nine articles evaluated the quality of care as part of their study. Choo et al surveyed 10 district hospitals in Ghana about quality improvement activities.[61] None of the hospitals surveyed reported an organised quality improvement programme (although this does not mean it was absent), and the only quality improvement activity related to surgery was found in one hospital and related to sterilisation of instruments.[61] Other authors echoed these findings by reporting a lack of clinical guidelines for surgery, anaesthesia and labour management[23 31 62] and a lack of staff trained in using them when they were available.[31]

To evaluate the quality of caesarean sections, Harfouche et al measured the 'decision to incision time' in a district hospital in Malawi and found that it reached '1.69 hours per caesarean section, far exceeding the recommended 30 min'.[38] They explained their findings only by a 'lack of operating theatre space and high patient volume' but also by an inefficient system to transfer patients to the operating theatre.[38]

Another major challenge to quality care in district hospitals in sub-Saharan Africa is the shortage of trained surgical and anaesthesia staff. Stewart et al[63] explained that shortage in staff resulted in unavailability of service during nights and weekends. Choo et al found that medical officers in Ghana had limited exposure to surgery during their training.[24] Harfouche et al reported that while clinical officers performed most caesarean sections at district level in Malawi, they were not sufficiently trained in performing emergency hysterectomies.[38] Positive outcomes were observed in cases where trained staff were present: Hall et al reported an 'association between the presence of Obstetricians and Gynaecologists in district hospitals in Ghana and an increase in evidence-based maternal care practices as well as moderate improvement in maternal mortality and stillbirth rates'.[37]

Some authors suggested interventions to improve the quality of care. Van den Akker et al suggested audit as a 'simple intervention requiring little technology' that is able to improve quality of care.[64] They reported a decrease in 68% of uterine ruptures in a district hospital in Malawi after implementing recommendations inferred from an audit. Recommendations included training of staff, better documentation and involvement of hospital management.[64]

Other authors insisted on the need for monitoring safety and quality of care in district hospitals as a first step toward improving quality.[40 61]

However, Smiley et al reported care of exceptional quality in a '63 bed district hospital' named the 'Volta River Authority Hospital' in Ghana.[48] In their study, they surveyed staff and patients about their perception of quality of care and conducted process mapping of surgical and obstetric care. They found that 'over 80% of employees across a variety of perioperative roles held a positive view of teamwork and safety climate within the institution' and 'surgical patients who were surveyed gave similarly positive indications of overall satisfaction'.[48] The perception of staff and patients was complemented by the observations of researchers during the process mapping. The WHO safety checklist was used in all operations witnessed and the authors reported the existence of a hospital wide morbidity and mortality conference as well as a safety training for new staff.[48]

### Surgical outcomes

The most commonly used indicators of outcome were perioperative complications and perioperative mortality. Fifteen articles reported information about perioperative complications,[14–17 25 33 35 38 39 41 43 62 64–66] the most reported being surgical site infection. Fifteen articles included information about perioperative mortality[10–12 15 16 25 26 30–32 35 37 49 61 62]; the most reported causes of death were sepsis, haemorrhage and anaesthesia complications.

In surgical obstetric care, the most reported complications were ruptured uterus, postpartum haemorrhage and wound dehiscence.[38 39 64] The most reported cause of maternal mortality was ruptured uterus.[15 18 25 26 30 32 35 61]

## DISCUSSION

Although it varies by country, our review found that district hospitals provide mainly emergency obstetric and general surgery and, in some cases, elective surgery covering a wide range of surgical specialties. The literature reports that, most often, surgical procedures are done under general or spinal anaesthesia.

Many of the procedures performed are by trained non-physician clinicians and medical officers. There is a paucity of data regarding quality of surgical care, surgical volume and perioperative morbidity and mortality.

Our study is a snapshot of two decades of published surgical activity in district hospitals. We had aimed to

compare it with other similar studies, but we found that none was appropriate, either in terms of similar methodology, scope or research question. We suggest that there is a challenge and an imperative for district hospitals to collect and publish information on key indicators so that standards can be measured, and quality improvement can be encouraged. Our preference is to start with the agreed indicators recommended by the Lancet Commission on Global Surgery.[6]

Based on the findings of this study, our recommendation is for Ministries of Health in sub-Saharan Africa to consider policies and resources that prioritise quality improvement practices for district-level surgical care in the region, including standardisation and quality improvement in surgical information systems.

We have reflected on the strengths and limitations of this study. When we designed the study protocol, we followed the WHO terminology regarding 'district hospitals'.[28] If we had used the term 'rural' in our search strategy, it would have added confusion as it includes a variety of structures, many of which are health centres or dispensaries without in-patient facilities. However, we could have missed relevant literature that did not specifically use the terminology 'district hospital' but instead used other terms such as 'rural hospital'. Our search strategy used the term 'Africa south of the Sahara' and this may have omitted studies that mentioned a country name but not the term 'Africa or 'south of the Sahara'. Our search included English, French and Portuguese texts but work in other languages may have been omitted. We sought to minimise subjectivity; therefore, two authors independently conducted the article selection and data extraction. Our study is prone to some publication bias as it is based only on information present in the published literature, and findings those were never published will have been omitted. It is unlikely that institutions those are not actively auditing activity would present it for publication, and even those district hospitals that do collect information of relevance to this study are unlikely to submit it for publication.[60] Indeed, these institutions face several barriers to publish, such as the cost of submissions to journals and overwhelming clinical demands reduce the time available for academic work. Furthermore, there is usually a cost of running trials/data collection and, if the research is published, there is often a pay wall burden resulting in a limited distribution of results for the readership in LMICs. Finally, our results are limited by the heterogeneity in how articles reported indicators such as personnel training level, surgical volume (in terms of minor and major operations), complications, outcomes and perioperative mortality. Some articles used surgical indicators, but these were usually nonstandard, and the core surgical indicators (geographical access to surgical care within 2 hours, surgical providers' density, total operative volume, in-hospital postoperative mortality and impoverishing and catastrophic cost burden) recommended by the Lancet Commission on Global Surgery[6] were not routinely used. However, this is the first review to systematically assess the literature for the scope, volume and quality of surgery in district hospitals in sub-Saharan Africa, to our knowledge.

## CONCLUSION

District hospitals play a significant role in surgical care in sub-Saharan African countries. This scoping review has provided a broad but incomplete data set on the current provision of surgical care in district hospitals in sub-Saharan Africa as reported in published studies. Based on the findings of this study, we recommend to researchers to conduct more primary surgical systems research in district hospitals to report on surgical activity, using core indicators suggested by the Lancet Commission on Global Surgery that will improve comparability.

**Contributors** ZB: investigation, data curation, methodology, formal analysis, writing-original draft, writing-review and editing. SS-A: investigation, formal analysis, writing- original draft, writing- review and editing. GD: project administration, investigation, writing-review and editing. CL: conceptualisation, supervision, writing-review and editing.

**Funding** This work was supported by SURG Africa through the European Union Horizon 2020 programme Grant Agreement number 733391. This work was also supported by the NIHR Biomedical Research Centre, Oxford, UK.

**Competing interests** None declared.

**Patient consent for publication** Not required.

**Provenance and peer review** Not commissioned; externally peer reviewed.

**Data availability statement** Data are available upon reasonable request.

**ORCID iDs**
Zineb Bentounsi http://orcid.org/0000-0002-9325-1954
Chris Lavy http://orcid.org/0000-0001-8794-3789

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
