## [Reviewer comments · BMJ Open]

ARTICLE DETAILS

TITLE (PROVISIONAL)	Surgical care in District Hospitals in sub-Saharan Africa: a Scoping Review
AUTHORS	Bentounsi, Zineb; Sheik-Ali, Sharaf; Drury, Grace; Lavy, Chris

VERSION 1 – REVIEW

REVIEWER	Rachael Morley University of Bristol North Bristol NHS Trust
REVIEW RETURNED	07-Aug-2020

GENERAL COMMENTS	It is clear that a vast amount of work and effort has gone into this paper, however there are some major issues that make it difficult to recommend publication. There is not a well defined reason for performing this scoping review laid out at the beginning of the paper. This should be beyond 'to summarise the literature.' The Lancet Commission on Global Surgery is mentioned at the very end, this may benefit from being discussed at the beginning if appropriate. The originality of the work is questionable, and seems to describe that which is already commonly known, or may be confirmed by a better method e.g. survey of relevant hospitals The introduction appears to discuss results. The final paragraph of the introduction mentions an aim of suggesting targets for policy, which are not mentioned later in the paper. The discussion needs a dedicated title and a complete rewrite in line with standard expectations, some of this is already included in the paper. -Comparison to other studies-Strengths-Weaknesses-Impact and clinical relevance-Further work-Conclusions
--

REVIEWER	Praveen Paul Rajaguru Warren Alpert Medical School of Brown University USA
REVIEW RETURNED	19-Aug-2020

GENERAL COMMENTS

Thank you for your contribution to the journal. This article fills an important gap in the literature by synthesizing information regarding district hospitals and their role in addressing surgical burden in SSA. Please see below for my comments and suggested revisions.

1) What was the rationale behind the use of the utilized search terminology? By limiting the search to purely "district hospitals," there appears to be the possibility that you may miss other similar hospitals providing a similar level of local care (i.e. regional hospitals, primary medical centers, etc.). What is the justification for this? The authors discuss this in the limitations section without much justification.

2) Per PRISMA guidelines, please include the GoogleScholar and Cochrane queries and findings in the PRISMA diagram. These can be noted as duplicates, but should be reported.

3) The study selection section may be written slightly clearer. It appears that the only inclusion criteria is whether the setting was "as district hospital in sub-Saharan Africa." However, after this sentence there is a description of the outcomes – were there outcomes used to determine inclusion? If not, please clarify.

4) What is the rationale for excluding dental and particularly ophthalmic surgical procedures? Please justify this.

5) The "charting the data" section may require some more details, and perhaps the charting form may be attached as a supplement. If possible, please include the different extraction items and fields.

6) The "collating, summarizing, and reporting the results" section may be clarified by listing the specific outcomes that are being numerically summarized. Further details on the thematic analysis may also be worth adding.

7) In the PRISMA diagram, please give the numerical breakdown along with the listed reasons for study exclusion. The diagram may also be formatted to avoid diagonal lines.

8) I recommend that designate the "Supplementary Table 1" should not be supplementary, and should simply be the "Table 1". Under study characteristics, especially since as the details of the articles are currently placed in a Supplementary Table, please describe the article characteristics in more detail beyond the usage of terms such as "most" or "a few", perhaps adding percentages may be useful in summarizing these findings.

9) In lines 54-56 of page 4, how did you assess that Ghana, Malawi, and Rwanda were most represented because of collaborations with HIC? Is this due to information that was gathered by the authors but not reported?

10) Please clarify and specify the section regarding "characteristics of district hospitals," perhaps presenting this information in a table would be beneficial.

11) For figures 2 and 3, please label the axes. Also this presentation of the data makes it unclear how many unique articles looked at each of the conditions. I recommend the usage

	of tables to make the numbers clearer for readers. It may be worth considering consolidating these findings into one table. 12) The first paragraph of the surgical volume section may be phrased more clearly and formally. 13) In general, separating the results from the discussion may allow the authors to more clearly and thoroughly present both the results and the discussion. I recommend looking into restructuring the paper in this way, if possible. For example, section 8.2 appears to be mainly results with no discussion. 14) It may be beneficial to add country breakdowns into Table 5 and Table 6. 15) The data behind section 8.1 of the results is not fully presented in the paper, perhaps a figure or table may be beneficial. This section could be more clearly and tightly structured (i.e. the paragraph discussing the Harfouche paper does not have much context). It may also be beneficial to give examples of the metrics used for "quality" in the studies that discussed quality in the same manner that you do in the beginning of section 8.2. 16) Please utilize the conclusion section to broaden the discussion, relevance, and future directions of the paper, rather than to primarily summarize the findings again. 17) Please include a data sharing statement and ensure full adherence to journal author guidelines. Based on this initial review of the manuscript, I would like to reiterate the importance of this work and applaud the contribution of the authors. However, I recommend major revisions to the paper prior to consideration for publication. The strongest recommendation is to clarify the presentation of methods and data. There is room to consolidate the existing manuscript and also provide more necessary details, as the manuscript length is 2768 and the journal limit is 4000. There should be more than enough room to clarify the methods in particular. Opportunities to further improve clarity and consistency in this manuscript may extend beyond the comments listed here. As such, the authors may benefit from also evaluating methods of further clarifying this paper.
--	--

REVIEWER	Mats Jan Lucas van der Wee Alrijne Healthcare Group/Alrijne Hospital Leiderdorp, the Netherlands.
REVIEW RETURNED	24-Aug-2020

GENERAL COMMENTS	The authors have performed a scoping review on the provision of surgical and anaesthesia care in district hospitals in sub-Saharan Africa. It is the first review to provide an extensive summary of the available literature regarding the size and structure of district hospitals in SSA and the type, number of operations and providers of surgical and anaesthesia care. They find that district hospitals play a fundamental role in improving access to timely surgical care in SSA, providing a wide scope of surgical operations. However, surgical care is provided by a heterogenous pool of often non-physician clinicians (depending on country), and there is a lack of quality control or quality of care is not reported in the literature.
---

	I appreciate the thorough review of the literature performed by the authors following the methodology of Arksey and O'Malley, which makes for a clearly structured analysis. Furthermore, the search strategy included not only articles in English, but also articles in French and Portuguese, and article screening was performed independently by two authors, contributing to the strength of the analysis. I would suggest a minor revision with small adjustments/explanations to tables and a more elaborate discussion regarding targets for further development of district hospitals in SSA. Specific comments: Results section: Except for "Surgical and anaesthesia providers", results are presented per article. Since the authors attempt to give an overview of the surgical care in a specific region (SSA), would it not be clearer to present the results per country rather than per article? I would suggest this in particular for table 2 and table 3, because it would give a clearer picture of the scope and volume of surgical care in the various countries in SSA. In addition, it may be interesting to see how the provision of surgical care differs across various countries in SSA, especially to identify specific targets for development and further research. Page 2, lines 26-27: The WHO defines a DALY as "the loss of the equivalent of one year of full health", and quantifies DALYs as DALYs per 1000 population. Since a DALY is already defined as "loss of health", why have you quantified the burden of disease as "... 38 disability adjusted life years (DALYs) lost per 1,000 population"? Shouldn't this be changed to "38 DALYs per 1,000 population"? Page 3, lines 41-42: Why was the search limited to references after January 1st, 2000? Is there any literature on surgical care in SSA available before that time? Page 3, line 57: Why are ophthalmic procedures not included? Page 4, lines 33-34: Since "Supplementary table 1" presents an overview of the countries in which the included articles were performed, could you provide this in-text instead of as supplementary material? Page 5, line 19: Table 1 reports 2 articles from sub-saharan Africa, however, all the other papers are also from Sub-Saharan African countries? Could you clarify this? Page 7, Table 3: As mentioned before, if possible, I would suggest structuring the results per country, or as the authors have done in "Supplementary Table 1", include countries in a column "Study setting". If possible, you could also include "Supplementary Table 1" in the main text under the Results section. Page 7, Table 3: How can the number of operations per year be estimated for articles that did not report a total number of operations?
--	--

	Page 11, lines 14-19: In the introduction the authors state that, following the overview presented in the review, they aim to suggest targets for policy and further development for surgical in the SSA region. However, only give suggestions with regards to further research in this area are given. Are there any targets for development/improvement of surgical care in SSA following from the literature discussed in your review? E.g. should SSA district hospitals adopt a structured model of care (with a trained surgeon or clinician, standardized structure etc.)? Should more countries in SSA use physicians instead of non-physician clinicians? What about monitoring of quality of care? Are there district hospitals in particular countries doing better than others and can we learn something from them?
--	---

VERSION 1 – AUTHOR RESPONSE

Reviewer: 1

Reviewer Name: Rachael Morley

Institution and Country: University of Bristol, North Bristol NHS Trust, UK

Please state any competing interests or state 'None declared': None

It is clear that a vast amount of work and effort has gone into this paper, however there are some major issues that make it difficult to recommend publication.

There is not a well defined reason for performing this scoping review laid out at the beginning of the paper. This should be beyond 'to summarise the literature.' The Lancet Commission on Global Surgery is mentioned at the very end, this may benefit from being discussed at the beginning if appropriate.

We have now mentioned this at the beginning.

The originality of the work is questionable, and seems to describe that which is already commonly known, or may be confirmed by a better method e.g. survey of relevant hospitals

The work is original. We agree that a standard survey may well inform this work in the future and we suggest that Lancet Commission indicators would play a role in standardising this.

The introduction appears to discuss results.

We apologise and have amended this.

The final paragraph of the introduction mentions an aim of suggesting targets for policy, which are not mentioned later in the paper.

We have amended this.

The discussion needs a dedicated title and a complete rewrite in line with standard expectations, some of this is already included in the paper.

-Comparison to other studies

-Strengths

-Weaknesses

- Impact and clinical relevance
- Further work
- Conclusions

Reviewer: 2

Reviewer Name: Praveen Paul Rajaguru

Institution and Country: Warren Alpert Medical School of Brown University, USA

Please state any competing interests or state 'None declared': None declared

Thank you for your contribution to the journal. This article fills an important gap in the literature by synthesizing information regarding district hospitals and their role in addressing surgical burden in SSA. Please see below for my comments and suggested revisions.

1) What was the rationale behind the use of the utilized search terminology? By limiting the search to purely "district hospitals," there appears to be the possibility that you may miss other similar hospitals providing a similar level of local care (i.e. regional hospitals, primary medical centers, etc.). What is the justification for this? The authors discuss this in the limitations section without much justification.

We used the term "District hospital" as it is commonly used in global surgical discussion and comment. Our study was done in order to see what published literature is available to inform on district hospital surgery. We conclude that the term may need expansion, and also that published literature is not the sole or even best source for information about district hospital surgical activity.

2) Per PRISMA guidelines, please include the GoogleScholar and Cochrane queries and findings in the PRISMA diagram. These can be noted as duplicates, but should be reported.

We have amended the methods section.

3) The study selection section may be written slightly clearer. It appears that the only inclusion criteria is whether the setting was "as district hospital in sub-Saharan Africa." However, after this sentence there is a description of the outcomes – were there outcomes used to determine inclusion? If not, please clarify.

Indeed, the outcomes were used to determine inclusion, we edited the text to make it clearer.

4) What is the rationale for excluding dental and particularly ophthalmic surgical procedures? Please justify this.

It is not uncommon for studies on global surgical activity to exclude these procedures. The reason is that they are often done as outpatients under local anaesthesia thus they do not get reported in surgical data from general operating theatres. They also tend to be performed by a cadre who are not involved in the main general surgical work of a DH.

5) The "charting the data" section may require some more details, and perhaps the charting form may be attached as a supplement. If possible, please include the different extraction items and fields.

Thank you for your suggestion. The data extraction form has been attached as a supplement and more details have been provided in the text.

6) The "collating, summarizing, and reporting the results" section may be clarified by listing the specific outcomes that are being numerically summarized. Further details on the thematic analysis may also be worth adding.

Thank you for your comment. Clarifications have been added to the text.

7) In the PRISMA diagram, please give the numerical breakdown along with the listed reasons for study exclusion. The diagram may also be formatted to avoid diagonal lines. The PRISMA diagram has been updated to reflect changes.

8) I recommend that designate the "Supplementary Table 1" should not be supplementary and should simply be the "Table 1". Under study characteristics, especially since as the details of the articles are currently placed in a Supplementary Table, please describe the article characteristics in more detail beyond the usage of terms such as "most" or "a few", perhaps adding percentages may be useful in summarizing these findings.

Thank you for your suggestion. We have made the changes that you have recommended.

9) In lines 54-56 of page 4, how did you assess that Ghana, Malawi, and Rwanda were most represented because of collaborations with HIC? Is this due to information that was gathered by the authors but not reported?

Indeed, it was due to information gathered but not reported. We didn't feel comfortable naming the collaborations, but with the information present on Table 1, the reader could explore the articles regarding these countries and come to the same conclusions.

10) Please clarify and specify the section regarding "characteristics of district hospitals," perhaps presenting this information in a table would be beneficial.

Thank you for your comment. A table with the number of beds per district hospital has been created.

11) For figures 2 and 3, please label the axes. Also this presentation of the data makes it unclear how many unique articles looked at each of the conditions. I recommend the usage of tables to make the numbers clearer for readers. It may be worth considering consolidating these findings into one table.

Thank you very much for this comment. We have deleted the figures and presented everything on the same table as you have recommended.

12) The first paragraph of the surgical volume section may be phrased more clearly and formally. Thank you, we have edited the paragraph.

13) In general, separating the results from the discussion may allow the authors to more clearly and thoroughly present both the results and the discussion. I recommend looking into restructuring the paper in this way, if possible. For example, section 8.2 appears to be mainly results with no discussion.

We have separated the results from the discussion in our resubmission.

14) It may be beneficial to add country breakdowns into Table 5 and Table 6.

Thanks for the suggestion. We fear that doing so will make the table less readable and we have given the country breakdown in the text.

15) The data behind section 8.1 of the results is not fully presented in the paper, perhaps a figure or table may be beneficial. This section could be more clearly and tightly structured (i.e. the paragraph discussing the Harfouche paper does not have much context). It may also be beneficial to give

examples of the metrics used for "quality" in the studies that discussed quality in the same manner that you do in the beginning of section 8.2.

16) Please utilize the conclusion section to broaden the discussion, relevance, and future directions of the paper, rather than to primarily summarize the findings again.

Thank you we have done this.

17) Please include a data sharing statement and ensure full adherence to journal author guidelines.

This has been done

Based on this initial review of the manuscript, I would like to reiterate the importance of this work and applaud the contribution of the authors. However, I recommend major revisions to the paper prior to consideration for publication. The strongest recommendation is to clarify the presentation of methods and data. There is room to consolidate the existing manuscript and also provide more necessary details, as the manuscript length is 2768 and the journal limit is 4000. There should be more than enough room to clarify the methods in particular. Opportunities to further improve clarity and consistency in this manuscript may extend beyond the comments listed here. As such, the authors may benefit from also evaluating methods of further clarifying this paper.

Reviewer: 3

Reviewer Name: Mats Jan Lucas van der Wee

Institution and Country: Alrijne Healthcare Group/Alrijne Hospital Leiderdorp, the Netherlands.

Please state any competing interests or state 'None declared': None declared

The authors have performed a scoping review on the provision of surgical and anaesthesia care in district hospitals in sub-Saharan Africa. It is the first review to provide an extensive summary of the available literature regarding the size and structure of district hospitals in SSA and the type, number of operations and providers of surgical and anaesthesia care. They find that district hospitals play a fundamental role in improving access to timely surgical care in SSA, providing a wide scope of surgical operations. However, surgical care is provided by a heterogenous pool of often non-physician clinicians (depending on country), and there is a lack of quality control or quality of care is not reported in the literature.

I appreciate the thorough review of the literature performed by the authors following the methodology of Arksey and O'Malley, which makes for a clearly structured analysis. Furthermore, the search strategy included not only articles in English, but also articles in French and Portuguese, and article screening was performed independently by two authors, contributing to the strength of the analysis. I would suggest a minor revision with small adjustments/explanations to tables and a more elaborate discussion regarding targets for further development of district hospitals in SSA.

Specific comments:

Results section: Except for "Surgical and anaesthesia providers", results are presented per article. Since the authors attempt to give an overview of the surgical care in a specific region (SSA), would it not be clearer to present the results per country rather than per article? I would suggest this in particular for table 2 and table 3, because it would give a clearer picture of the scope and volume of surgical care in the various countries in SSA.

In addition, it may be interesting to see how the provision of surgical care differs across various countries in SSA, especially to identify specific targets for development and further research.

Thank you for your comment. We have amended table 3 (now table 5) to include the study setting. We will also add a table in supplementary material to present operations performed per country.

Page 2, lines 26-27: The WHO defines a DALY as “the loss of the equivalent of one year of full health”, and quantifies DALYs as DALYs per 1000 population. Since a DALY is already defined a ‘loss of health’, why have you quantified the burden of disease as “.. 38 disability adjusted life years (DALYs) lost per 1,000 population”? Shouldn’t this be changed to “38 DALYs per 1,000 population”?

We agree. This was a mistake. Thanks for helping us to correct it.

Page 3, lines 41-42: Why was the search limited to references after January 1st, 2000? Is there any literature on surgical care in SSA available before that time?

The search was limited to the year 2000 because we wanted to get a recent picture of the situation. We thought that comparing articles across several decades would not be appropriate.

Page 3, line 57: Why are ophthalmic procedures not included?

Page 4, lines 33-34: Since “Supplementary table 1” presents an overview of the countries in which the included articles were performed, could you provide this in-text instead of as supplementary material?

Yes, it has been included in text for the resubmission. Thanks

Page 5, line 19: Table 1 reports 2 articles from sub-saharan Africa, however, all the other papers are also from Sub-Saharan African countries? Could you clarify this?

This is a very good comment, thank you. We meant that these two articles presented information from a regional perspective and were not country specific. We made it clearer on the table.

Page 7, Table 3: As mentioned before, if possible, I would suggest structuring the results per country, or as the authors have done in “Supplementary Table 1”, include countries in a column “Study setting”.

If possible, you could also include “Supplementary Table 1” in the main text under the Results section.

We have made both changes. Thank you for the suggestion.

Page 7, Table 3: How can the number of operations per year be estimated for articles that did not report a total number of operations?

All articles included in the table reported a total number of operations.

Page 11, lines 14-19: In the introduction the authors state that, following the overview presented in the review, they aim to suggest targets for policy and further development for surgical in the SSA region. However, only give suggestions with regards to further research in this area are given. Are there any targets for development/improvement of surgical care in SSA following from the literature discussed in your review? E.g. should SSA district hospitals adopt a structured model of care (with a trained surgeon or clinician, standardized structure etc.)? Should more countries in SSA use physicians instead of non-physician clinicians? What about monitoring of quality of care? Are there district hospitals in particular countries doing better than others and can we learn something

from them?

Thank you. We are grateful for this incisive comment. We had hoped that there would be significant data to inform policy and have made some comments on policy, but feel that overall, the key message is one of improving the quality of data available from DHs.

VERSION 2 – REVIEW

REVIEWER	Rachael Miller Centre for Surgical Research, University of Bristol
REVIEW RETURNED	20-Oct-2020

GENERAL COMMENTS	Thank you for responding to suggested changes, on the whole I think the research question, results and implications are clearer and it is improved because of this. However, there is still plenty of room for further improvement. Although you have mentioned the Lancet Commission on global surgery now in the introduction, it would benefit from being defined e.g. the Lancet commission, which considers X, Y and Z for A B and C, reinforces this emphasis... Active and passive voices are used interchangeably in the paper e.g. 'Articles were included' and 'we considered a systematic review.' Suggest choosing one and sticking to that throughout. Aims mentioned at the end of the introduction, which are more comprehensive than objective stated in abstract, should be included in the objectives heading of the abstract. p4 line 29 - should it be 'thus'? Perhaps add in something about this making it more relevant to current practice p5 line 22 - suggest removing 'results and discussion are simultaneously presented' Table 1. - suggest references come after author name rather than in a separate column Whilst I appreciate the now organised discussion, the subheadings should be removed and each paragraph should correspond to each section. Limitations should be combined and 'however this study has some limitations' should come at the beginning of the limitations paragraph not the end of the strengths. p18 line 18 - could you expand on what the indicators are? This may be more relevant in the introduction where it is first mentioned You refer to district hospitals throughout the paper and then suddenly use the acronym DH in the discussion, suggest keeping the wording in full throughout and just don't define it as DH (as you have done in the introduction)
--

	I look forward to reading a further revision. To make responses easier to read through, could you please copy any edited/newly inserted text into the response to authors document.
--	---

REVIEWER	Praveen Paul Rajaguru Warren Alpert Medical School of Brown University, USA
REVIEW RETURNED	23-Oct-2020

GENERAL COMMENTS	I would like to again thank the authors for this important contribution to the literature. This piece fills an important gap and does a good job of justifying the importance of looking at district hospitals in SSA, and the new revision has addressed many of the initial comments. Please see below for new comments and suggestions: Introduction: 1) It may not be necessary to include the background on p.3 lines 27-29 in this manner. While interesting, the discussion of the author connection to the paper subject may not be necessary in this manuscript. Methods: 1) Please give a rationale for limiting the study search to January 1st 2000 onwards, why were earlier studies not included? 2) If excluding dental and ophthalmic procedures on the basis of the majority being outpatient, it would appear that studies focusing on outpatient procedures are an exclusion criteria and should be listed as such. Results: 1) Given the number of urological procedures listed (hydrocele, urethrotomy, bladder stone) it may be beneficial to separate urology from the category of general surgery. 2) P. 13 Lines 29-32 may better belong in the methods section. 3) Were there any differences in anaesthesia modality based on specialty, location, or hospital volume? This might be worth including in the discussion on anaesthesia. 4) Were there differences in surgical outcomes based on specialty, location or hospital volume? This may also be worth including; it appears there is a focus on obstetrics in the discussion, but were there other specialties reporting complications? Discussion: 1) If it is possible that hospitals such as "rural hospitals" may have been missed per p. 17 lines 50-52, why was the language not broadened in the initial search? Especially given that fewer than 400 papers were found in the initial search, it appears that the initial search may have been rather narrow. Is there a justification for this? 2) Regarding the "impact for future research" (P. 18 Lines 14-19) it may be useful to address some of the limitations as to why DHs have difficulty in publishing their research. Publication costs for many journals may be high, there may be lower prioritization of academic research based on increased clinical demands, etc. It
--

	would be beneficial to understand what the barriers DHs may have to publishing their data in this manuscript. 3) On "Impact for policy" P. 18 lines 23-25, it appears unfair to assume that DHs are not recording their surgical activity. As you mentioned, there is publication bias, but lack of availability of data in the literature is not equivalent to a lack of recording. This statement may need to be rephrased or qualified. Conclusion: 1) At some point in the conclusion, please make sure to note that this is "based on the findings of this study," this will provide a good tie in for the ending.
--	--

REVIEWER	Mats Jan Lucas van der Wee Alrijne Healthcare Group/Alrijne Hospital Leiderdorp, the Netherlands
REVIEW RETURNED	06-Nov-2020

GENERAL COMMENTS	I think the authors have improved their review following the suggestions made by the reviewers. I feel like the authors have been able to adequately address and answer my comments and suggestions. The addition of (previously supplementary) Table 1 and the inclusion of the column "Study setting" to Table 1 and Table 5 provide a much clearer overview of the scope of surgical care across the different countries in SSA, and thus make for an improved paper. In addition, the data extraction has been clarified by explaining and providing the data extraction form. The discussion is now clearly structured following standard topics addressed in discussion sections. Furthermore, the authors have elaborated on strengths and limitations of the paper and the implications for further research and development. However, I think some key points in the Discussion section are still missing. Hence, I would suggest a minor revision. Find my specific comments below: Page 4, line 33: "...aiming to provide a general overview of the reported current surgical capacity and delivery, advance current knowledge and suggest targets for further development and research within the region of SSA." I assume that the rationale for the authors to perform the study is to provide a general overview of surgical care in DHs in SSA, providing background information which can be used to advance current knowledge and identify targets for further development. Hence, I would suggest amending the sentence to: "...provide a general overview of the reported current surgical capacity and delivery, in order to advance current knowledge..". Page 5, line 13: "We limited the search to references published after January 1st, 2000 this giving us a 20 year study period." In the reply to my comment on the original submission, you explained that the reason for this limitation was that you wanted to get a recent picture of the situation, which is a good reason for limiting the search. However, in the revised paper, this is not explained. I think "giving us a 20 year study period" is obvious information and can be left out. I suggest you would amend this in the text; "... after January 1st, 2000, because we wanted to get a recent picture of the situation".
---

	Page 18-19, Discussion:  - No comparison to other studies was made because no appropriate similar studies were found. If so, why were these studies not appropriate? - In the introduction, you have now excluded targets for further policy as an aim of the review. Subsequently, in the discussion, you mention that it is not the role of the paper to influence or determine policy. Why would it not be the role of the paper to suggest (specific) targets for further policy? You already point out that the findings of the paper indicate areas for improvement, such as improving quality of data and data recording in DHs in sub-Saharan Africa. In my opinion, this does not only translate to impact on future research, but also on policy: if information regarding key indicators for surgical activity and anesthesia (such as recommended by the Lancet Commission on Global Surgery) needs to be collected in DHs, wouldn't policy need to change in order to ensure improvement of quality of data collected and provided by DHs, so that this data can be used for future research?
--	---

VERSION 2 – AUTHOR RESPONSE

Reviewer: 1

Reviewer Name: Rachael Miller

Institution and Country: Centre for Surgical Research, University of Bristol

Please state any competing interests or state 'None declared': None declared

Comments to the Author

Thank you for responding to suggested changes, on the whole I think the research question, results and implications are clearer and it is improved because of this. However, there is still plenty of room for further improvement.

Thank you for your helpful feedback.

- Although you have mentioned the Lancet Commission on global surgery now in the introduction, it would benefit from being defined e.g. the Lancet commission, which considers X, Y and Z for A B and C, reinforces this emphasis...

This has been revised to: 'The Lancet Commission on Global Surgery (65), which defines scalable solutions for the provision of quality surgical and anaesthesia care for all, reinforces this emphasis on district hospitals and echoes other authors' calls to consider district hospitals as a central component in strengthening surgical care in SSA'

- Active and passive voices are used interchangeably in the paper e.g. 'Articles were included' and 'we considered a systematic review.' Suggest choosing one and sticking to that throughout. This has been standardized as far as possible.

- Aims mentioned at the end of the introduction, which are more comprehensive than objective stated in abstract, should be included in the objectives heading of the abstract. Thank you for your suggestion. This has been revised to: "To provide a general overview of the reported current surgical capacity and delivery in order to advance current knowledge and suggest targets for further development and research within the region of SSA"

- p4 line 29 - should it be 'thus'? Perhaps add in something about this making it more relevant to current practice This has been rephrased to: "Indeed, these institutions face several barriers to publish, such as the cost of submissions to journals or overwhelming clinical demands that reduces the time available for academic work. Furthermore, there is usually a cost of running trials/data collection and, if the research is published, there is often a pay wall burden resulting in a limited distribution of results for the readership in LMIC".

- p5 line 22 - suggest removing 'results and discussion are simultaneously presented' This has been amended.

- Table 1. - suggest references come after author name rather than in a separate column. This has been amended.

- Whilst I appreciate the now organised discussion, the subheadings should be removed and each paragraph should correspond to each section. This has been amended.

- Limitations should be combined and 'however this study has some limitations' should come at the beginning of the limitations paragraph not the end of the strengths. We have made these amendments.

- p18 line 18 - could you expand on what the indicators are? This may be more relevant in the introduction where it is first mentioned.

This has been revised to: 'Some articles used surgical indicators but these were usually non-standard, and the core surgical indicators (geographical access to surgical care within 2 hours, surgical providers density, total operative volume, in hospital postoperative mortality, impoverishing and catastrophic cost burden)recommended by the Lancet Commission on Global Surgery(65) were not routinely used.'

- You refer to district hospitals throughout the paper and then suddenly use the acronym DH in the discussion, suggest keeping the wording in full throughout and just don't define it as DH (as you have done in the introduction). We have amended this.

Reviewer: 2

Reviewer Name: Praveen Paul Rajaguru

Institution and Country: Warren Alpert Medical School of Brown University, USA

Please state any competing interests or state 'None declared': None declared

Comments to the Author

I would like to again thank the authors for this important contribution to the literature. This piece fills an important gap and does a good job of justifying the importance of looking at district hospitals in SSA, and the new revision has addressed many of the initial comments. Please see below for new comments and suggestions:

Thank you for your helpful feedback.

Introduction:

1) It may not be necessary to include the background on p.3 lines 27-29 in this manner. While interesting, the discussion of the author connection to the paper subject may not be necessary in this manuscript. This has been deleted.

Methods:

1) Please give a rationale for limiting the study search to January 1st 2000 onwards, why were earlier

studies not included? We have made this clear in the text, also in line with reviewer 3's comment. We limited the search to references published after January 1st, 2000 in order to review more recent studies and gain a clearer understanding of more recent developments in SSA.

2) If excluding dental and ophthalmic procedures on the basis of the majority being outpatient, it would appear that studies focusing on outpatient procedures are an exclusion criteria and should be listed as such. We have amended this to: "We included all surgical procedures except those typically done in outpatient departments, such as dental and ophthalmic procedures."

Results:

1) Given the number of urological procedures listed (hydrocele, urethrotomy, bladder stone) it may be beneficial to separate urology from the category of general surgery. This is a good point and we have thought about it. However, in the context of our study: district hospitals in sub-Saharan Africa, surgical practice is less specialised and therefore these procedures come under providers of general surgery. We wanted our study to reflect the practice of the context.

2) P. 13 Lines 29-32 may better belong in the methods section. Thank you for your suggestion, however, we prefer to leave it where it is for clarity.

3) Were there any differences in anaesthesia modality based on specialty, location, or hospital volume? This might be worth including in the discussion on anaesthesia. We have not analysed this in our study.

4) Were there differences in surgical outcomes based on specialty, location or hospital volume? This has not been analysed

This may also be worth including; it appears there is a focus on obstetrics in the discussion, but were there other specialties reporting complications? The apparent focus on obstetrics might be explained by the volume of publications on the topic which resulted in a volume of material for analysis, this was not the case of any other speciality in our selection of papers.

Discussion:

1) If it is possible that hospitals such as "rural hospitals" may have been missed per p. 17 lines 50-52, why was the language not broadened in the initial search? Especially given that fewer than 400 papers were found in the initial search, it appears that the initial search may have been rather narrow. Is there a justification for this?

When we designed the study protocol, we followed the WHO terminology regarding "district hospitals" https://www.who.int/maternal_child_adolescent/documents/9241545755/en/. The use of the term "rural" would have added confusion as it includes a variety of structures, many of which are health centres or dispensaries without in-patient facilities.

2) Regarding the "impact for future research" (P. 18 Lines 14-19) it may be useful to address some of the limitations as to why DHs have difficulty in publishing their research. Publication costs for many journals may be high, there may be lower prioritization of academic research based on increased clinical demands, etc. It would be beneficial to understand what the barriers DHs may have to publishing their data in this manuscript.

Thank you very much for this suggestion, we have made edits to the text: "Our study is a snapshot of two decades of published surgical activity in district hospitals. We had aimed to compare it with other similar studies, but we found none that were appropriate, either in terms of similar methodology, or scope and research question. We suggest that there is a challenge and an imperative for district hospitals to collect information on key indicators so that standards can be measured, and quality improvement encouraged. Our preference is to start with the agreed indicators recommended by the

Lancet Commission on Global Surgery. (65)”

3) On "Impact for policy" P. 18 lines 23-25, it appears unfair to assume that DHs are not recording their surgical activity. As you mentioned, there is publication bias, but lack of availability of data in the literature is not equivalent to a lack of recording. This statement may need to be rephrased or qualified.

Thanks again for this suggestion, we have amended this to: “Based on the findings of this study we suggest that the key policy recommendation for Ministries of Health in sub-Saharan Africa is to support district hospitals, through policy changes and resources, in standardizing and improving quality of recording of surgical activity, and quality improvement practice for district-level surgical care the region”.

Conclusion:

1) At some point in the conclusion, please make sure to note that this is "based on the findings of this study," this will provide a good tie in for the ending. We have now included revised text: ‘Based on the findings of this study, we recommend to researchers to conduct more primary surgical systems research in district hospitals to report on surgical activity, using core indicators suggested by the Lancet Commission on Global Surgery that will improve comparability.’

Reviewer: 3

Reviewer Name: Mats Jan Lucas van der Wee

Institution and Country: Alrijne Healthcare Group/Alrijne Hospital Leiderdorp, the Netherlands

Please state any competing interests or state ‘None declared’: None declared

Comments to the Author

I think the authors have improved their review following the suggestions made by the reviewers. I feel like the authors have been able to adequately address and answer my comments and suggestions. The addition of (previously supplementary) Table 1 and the inclusion of the column “Study setting” to Table 1 and Table 5 provide a much clearer overview of the scope of surgical care across the different countries in SSA, and thus make for an improved paper. In addition, the data extraction has been clarified by explaining and providing the data extraction form.

The discussion is now clearly structured following standard topics addressed in discussion sections. Furthermore, the authors have elaborated on strengths and limitations of the paper and the implications for further research and development. However, I think some key points in the Discussion section are still missing. Hence, I would suggest a minor revision.

Thank you for your helpful feedback.

Find my specific comments below:

Page 4, line 33: “..aiming to provide a general overview of the reported current surgical capacity and delivery, advance current knowledge and suggest targets for further development and research within the region of SSA.”

I assume that the rationale for the authors to perform the study is to provide a general overview of surgical care in DHs in SSA, providing background information which can be used to advance current knowledge and identify targets for further development.

Hence, I would suggest amending the sentence to: “..provide a general overview of the reported current surgical capacity and delivery, *in order to* advance current knowledge..”. Thank you, we have made this amendment: “aiming to provide a general overview of the reported current surgical capacity and delivery in order to advance current knowledge and suggest targets for further development and research within the region of SSA”

- Page 5, line 13: “We limited the search to references published after January 1st, 2000 this giving us a 20 year study period.”

In the reply to my comment on the original submission, you explained that the reason for this limitation was that you wanted to get a recent picture of the situation, which is a good reason for limiting the search. However, in the revised paper, this is not explained. I think “giving us a 20 year study period” is obvious information and can be left out.

I suggest you would amend this in the text; “... after January 1st, 2000, because we wanted to get a recent picture of the situation”.

We have revised this to: ‘We limited the search to references published after January 1st, 2000 because we wanted to have a recent picture of the situation.’

- Page 18-19, Discussion:

- No comparison to other studies was made because no appropriate similar studies were found. If so, why were these studies not appropriate?

We have revised the text to clarify this: ‘We had aimed to compare it with other similar studies, but we found none that was appropriate, either in terms of similar methodology or scope.’

- In the introduction, you have now excluded targets for further policy as an aim of the review. Subsequently, in the discussion, you mention that it is not the role of the paper to influence or determine policy. Why would it not be the role of the paper to suggest (specific) targets for further policy? You already point out that the findings of the paper indicate areas for improvement, such as improving quality of data and data recording in DHs in sub-Saharan Africa. In my opinion, this does not only translate to impact on future research, but also on policy: if information regarding key indicators for surgical activity and anesthesia (such as recommended by the Lancet Commission on Global Surgery) needs to be collected in DHs, wouldn’t policy need to change in order to ensure improvement of quality of data collected and provided by DHs, so that this data can be used for future research?

Thank you, we have revised the text to clarify the relevance of our findings for policymakers: “Based on the findings of this study we suggest that the key policy recommendation for Ministries of Health in sub-Saharan Africa is to support district hospitals, through policy changes and resources, in standardizing and improving quality of recording of surgical activity, and quality improvement practice for district-level surgical care the region”

VERSION 3 – REVIEW

REVIEWER	Praveen Paul Rajaguru Warren Alpert Medical School of Brown University
REVIEW RETURNED	01-Jan-2021

GENERAL COMMENTS	Thank you again for this contribution to the literature. While excellent improvements have been made, further changes are still required with a focus on improving clarity, nuance, and consistency. This piece has much to add to the literature, and further improvements can add to the utility of the paper. Please see below for my specific comments: Abstract: N/A Introduction: 1) P. 3 Line 15-16: This is a strong statement that "the main burden in several countries remains on non-physician clinicians to manage surgical cases." The cited paper for this statement (4, Galukande)
---

	does not mention this in the paper. Please revise this statement or support it with appropriate sources. 2) P. 3 Line 29-36: These lines compose one sentence; please break this up into several sentences for clarity. Methods: 1) P. 3 Line 51-57: The outcome "what types of surgical procedures" is listed as both a primary and secondary outcome; it should only be listed as one. 2) P. 4 Line 16-18: If Google Scholar and Cochrane were also searched, this should be included in the PRISMA flow diagram, even if duplicates were removed as with the other databases. Results: In general for this section, please try to give more detail and nuance in the text with better precision; it will be useful for readers as going through the tables in detail may be more difficult. The results should read succinctly but clearly summarize your findings. 1) Please list the number of articles in parentheses whenever percentages are referenced; this will give readers a clearer idea of how many articles fit the discussed categories. 2) P. 6 Line 52-54: The objective line here should read "Investigate anaesthesia capacity" 3) P. 6 Line 55-60: The objective line here should read "Explore provider perspectives" or "Explore providers' perspectives" 4) P. 9 Line 18-20: The objective line here should read "estimate the surgical volume," in keeping with the usage of verbs throughout the rest of the table 5) P. 10 Line 54-59: Giving specific numbers and percentages here may be useful for the readers 6) P. 11 Line 33-34: More nuance may be given for the line "district hospitals served rural or urban populations"; was one population served more frequently? How did certain hospitals serve both urban and rural populations simultaneously? 7) P. 11 Line 52-60: While the numbers and data listed here describe the number of articles that discuss a specific operation, it might be worth considering on aggregate how many procedures total were also performed across studies for each type of operation (i.e. "across all studies, c-section was the most commonly reported procedure, discussed in 30 articles comprising at least 3,234 total cases"). Although these numbers may not be available for every study, this may give a more accurate picture of the scope of procedures than solely summarizing representation within articles. 8) Please be consistent in the results section when referring readers to tables. At times, the Table # is listed in parentheses at the end of a sentence, and other times "as seen in tables" is used at the beginning of a sentence.
--	--

	9) P. 14 Line 45-46: As none of the published articles discussed hospitals with a volume <100, it may be beneficial to remove this row of table 6. 10) P. 14 Line 58-60: The modality of anaesthesia, while useful, may benefit from increased nuance and analysis. This is especially true given the variety of surgical procedures and settings observed in the results. A few sentences discussing when certain modalities are used should be strongly considered. 11) P. 15 Line 17-18: Although this sentence notes "there was no clear majority in terms of the number of articles mentioning each cadre," the above lines discuss and cite studies where the different cadres are mentioned. There are also numbers reported in Tables 7 and 8. Even if the cadres are only mentioned in (for example) 40 articles, majorities may still be stated of these 40 and should be listed. The same is true for the discussion of anaesthesia providers. 12) P. 17 Line 22-28: Timeliness of care is not necessarily equivalent to quality; per the Lancet modeling study, timeliness is a metric of access. https://www.thelancet.com/journals/langlo/article/PIIS2214-109X(15)70115-4/fulltext#:~:text=In%20a%20world%20focusing%20on,have%20access%20to%20surgical%20care. This section may require some rephrasing; perhaps framing this around "safety, quality, and timeliness" may be better? The authors may determine a clearer mode of presenting this. 13) P. 17 Line 9-11: The above section is focused on quality, and 8.2 appears to be focused on surgical outcomes. However, the opening of 8.2 also discusses indicators of quality; please make these sections consistent. Perhaps a reframing as discussed in previous comments may be beneficial. 14) P. 17 Line 19-20: This should read "maternal mortality." Discussion: 1) P. 17 Line 35-36: In 8.2, the most reported complication is SSI, the most reported causes of death were sepsis, haemorrhage, and anaesthesia complications, and the most reported obstetric complications were ruptured uterus, post-partum haemorrhage, and wound dehiscence. However, no rankings are given. As such, Lines 35-36 in the discussion are unclear; there appear to be many more common complications than just infections and post-partum haemorrhage. Please make this line consistent with the results and abstract. 2) P. 17 Line 45-46: Please include the justification for the first limitation (i.e. why the search language was not expanded) that was given in the previous "author response" here. 3) P. 17 Lines 47-55: Excellent explanation here, please cite relevant sources to back up these points. 4) P. 18 Line 12-14: Please make this line consistent with the discussion in P. 17 Lines 47-55, and note an "imperative for district hospitals to collect and publish information," as opposed to just "collect."
--	--

	5) P. 18 Line 19-23: This paragraph/sentence may be edited for clarity. Additional nuance may also be added to expand upon these very important recommendations. Conclusion: 1) These two paragraphs may be combined.
--	---

REVIEWER	MJL van der Wee Arijne Healthcare Group, the Netherlands
REVIEW RETURNED	24-Jan-2021

GENERAL COMMENTS	I appreciate the thorough revision the authors have performed, taking into account all the reviewers' comments. The review has clearly improved, especially the discussion section with regards to limitations and future policy. I would still advise a few minor adjustments. Methods p.4, line 20: I would suggest changing "in order ... in SSA' to because we wanted to provide a recent overview and explain that this makes studies before 2000 irrelevant. Results p. 11, line 42 (Characteristics of district hospitals): Elsewhere you have used the term "district hospitals" instead of the acronym DH, however, in this section you suddenly use DH again. Discussion: I would suggest some adjustments to the structure of the discussion:  - For a clearer structure, move the strengths and limitations paragraph to the end of the discussion, as the last paragraph. - p. 17, line 57: This sentence ("this is the first review... to our knowledge) should be at the end of the limitations section, starting with "However...", explaining the importance of the study despite its limitations.
--

VERSION 3 – AUTHOR RESPONSE

Reviewer: 2

Dr. Praveen Rajaguru, Brown University Warren Alpert Medical School

Comments to the Author:

Thank you again for this contribution to the literature. While excellent improvements have been made, further changes are still required with a focus on improving clarity, nuance, and consistency. This piece has much to add to the literature, and further improvements can add to the utility of the paper. Please see below for my specific comments:

Abstract:
N/A

Introduction:

1) P. 3 Line 15-16: This is a strong statement that "the main burden in several countries remains on non-physician clinicians to manage surgical cases." The cited paper for this statement (4, Galukande) does not mention this in the paper. Please revise this statement or support it with appropriate sources.

We have changed the reference for this statement.

2) P. 3 Line 29-36: These lines compose one sentence; please break this up into several sentences for clarity.

We have amended this to: 'We considered a systematic review of district hospital surgery, however there were a limited number of existing studies in this field and a wide range of methodology used. Therefore, we chose to change our review methodology and undertake a scoping review of surgical care delivery in district hospitals in sub-Saharan Africa. Our aim is to provide a general overview of the reported current surgical capacity and delivery in order to advance current knowledge and suggest targets for further development and research within the region of sub-Saharan Africa.'

Methods:

1) P. 3 Line 51-57: The outcome "what types of surgical procedures" is listed as both a primary and secondary outcome; it should only be listed as one.

This has been deleted from the secondary outcomes.

2) P. 4 Line 16-18: If Google Scholar and Cochrane were also searched, this should be included in the PRISMA flow diagram, even if duplicates were removed as with the other databases.

We have added Google Scholar and Cochrane to the PRISMA flow diagram.

Results:

In general for this section, please try to give more detail and nuance in the text with better precision; it will be useful for readers as going through the tables in detail may be more difficult. The results should read succinctly but clearly summarize your findings.

1) Please list the number of articles in parentheses whenever percentages are referenced; this will give readers a clearer idea of how many articles fit the discussed categories.

We have corrected this.

2) P. 6 Line 52-54: The objective line here should read "Investigate anaesthesia capacity"

We have corrected this.

3) P. 6 Line 55-60: The objective line here should read "Explore provider perspectives" or "Explore providers' perspectives"

We have corrected this to: 'Explore providers' perspectives on obstacles to surgery'

4) P. 9 Line 18-20: The objective line here should read "estimate the surgical volume," in keeping with the usage of verbs throughout the rest of the table

We have corrected this.

5) P. 10 Line 54-59: Giving specific numbers and percentages here may be useful for the readers

We have replaced with: 'Of the 52 articles included for analysis, covering 16 countries, 3 articles presented data from multiple countries(4,30,42) and 2 gave a regional perspective.(33,37)'

6) P. 11 Line 33-34: More nuance may be given for the line "district hospitals served rural or urban populations"; was one population served more frequently? How did certain hospitals serve both urban and rural populations simultaneously?

We have replaced with: 'District hospitals served rural or urban populations, with some serving both

groups (for example if they were situated on the edge of towns).'

7) P. 11 Line 52-60: While the numbers and data listed here describe the number of articles that discuss a specific operation, it might be worth considering on aggregate how many procedures total were also performed across studies for each type of operation (i.e. "across all studies, c-section was the most commonly reported procedure, discussed in 30 articles comprising at least 3,234 total cases"). Although these numbers may not be available for every study, this may give a more accurate picture of the scope of procedures than solely summarizing representation within articles. Thank you for raising this point, we had hoped to aggregate the procedures, however as the data was incomplete we have not included it.

8) Please be consistent in the results section when referring readers to tables. At times, the Table # is listed in parentheses at the end of a sentence, and other times "as seen in tables" is used at the beginning of a sentence. We have standardized the table references.

9) P. 14 Line 45-46: As none of the published articles discussed hospitals with a volume <100, it may be beneficial to remove this row of table 6. Thank you for this point, we felt that this row still represented a result therefore we have included it in the table. However, we are happy to remove it at the Editors' request.

10) P. 14 Line 58-60: The modality of anaesthesia, while useful, may benefit from increased nuance and analysis. This is especially true given the variety of surgical procedures and settings observed in the results. A few sentences discussing when certain modalities are used should be strongly considered. Thank you, we would have liked to comment on when certain modalities of anaesthesia were used, however the studies lacked some contextual detail and did not make it clear why certain types of anaesthesia were used.

11) P. 15 Line 17-18: Although this sentence notes "there was no clear majority in terms of the number of articles mentioning each cadre," the above lines discuss and cite studies where the different cadres are mentioned. There are also numbers reported in Tables 7 and 8. Even if the cadres are only mentioned in (for example) 40 articles, majorities may still be stated of these 40 and should be listed. The same is true for the discussion of anaesthesia providers. To avoid confusion we have deleted the sentence 'there was no clear majority in terms of the number of articles mentioning each cadre' as we think the detailed figures in the preceding sentence and Tables 7 and 8 are clear.

12) P. 17 Line 22-28: Timeliness of care is not necessarily equivalent to quality; per the Lancet modeling study, timeliness is a metric of access. [https://www.thelancet.com/journals/langlo/article/PIIS2214-109X\(15\)70115-4/fulltext#:~:text=In%20a%20world%20focusing%20on,have%20access%20to%20surgical%20care.](https://www.thelancet.com/journals/langlo/article/PIIS2214-109X(15)70115-4/fulltext#:~:text=In%20a%20world%20focusing%20on,have%20access%20to%20surgical%20care.) This section may require some rephrasing; perhaps framing this around "safety, quality, and timeliness" may be better? The authors may determine a clearer mode of presenting this. We agree that generally timeliness is a metric of access to care, however in this specific context of 'decision to incision time' for caesarean sections, we think that timeliness is an indicator of quality of care.

13) P. 17 Line 9-11: The above section is focused on quality, and 8.2 appears to be focused on surgical outcomes. However, the opening of 8.2 also discusses indicators of quality; please make these sections consistent. Perhaps a reframing as discussed in previous comments may be beneficial.

We have deleted 'and quality' so the opening sentence of 8.2 is now: 'The most commonly used indicators of outcome were peri-operative complications and peri-operative mortality.'

14) P. 17 Line 19-20: This should read "maternal mortality."
We have corrected this.

Discussion:

1) P. 17 Line 35-36: In 8.2, the most reported complication is SSI, the most reported causes of death were sepsis, haemorrhage, and anaesthesia complications, and the most reported obstetric complications were ruptured uterus, post-partum haemorrhage, and wound dehiscence. However, no rankings are given. As such, Lines 35-36 in the discussion are unclear; there appear to be many more common complications than just infections and post-partum haemorrhage. Please make this line consistent with the results and abstract.

We have deleted: 'However, the most reported complications are infections and post-partum haemorrhage.' From the discussion section, as the details of the reported complications are in the preceding few paragraphs.

2) P. 17 Line 45-46: Please include the justification for the first limitation (i.e. why the search language was not expanded) that was given in the previous "author response" here.

We have included the justification so that it now reads: 'We have reflected on the strengths and limitations of this study. When we designed the study protocol, we followed the WHO terminology regarding 'district hospitals.' If we had used the term 'rural' in our search strategy, it would have added confusion as it includes a variety of structures, many of which are health centres or dispensaries without in-patient facilities. However, we could have missed relevant literature that did not specifically use the terminology 'district hospital' but instead used other terms such as 'rural hospital'.'

3) P. 17 Lines 47-55: Excellent explanation here, please cite relevant sources to back up these points. We have added references here.

4) P. 18 Line 12-14: Please make this line consistent with the discussion in P. 17 Lines 47-55, and note an "imperative for district hospitals to collect and publish information," as opposed to just "collect."

We have amended this to: 'We suggest that there is a challenge and an imperative for district hospitals to collect and publish information...'

5) P. 18 Line 19-23: This paragraph/sentence may be edited for clarity. Additional nuance may also be added to expand upon these very important recommendations.

We have amended to: 'Based on the findings of this study, our recommendation is for Ministries of Health in sub-Saharan Africa to consider policies and resources that prioritize quality improvement practices for district-level surgical care in the region, including standardization and quality improvement in surgical information systems.'

Conclusion:

1) These two paragraphs may be combined.
We have combined these paragraphs.

Reviewer: 2

Competing interests of Reviewer: None declared

Reviewer: 3

Mr. Mats van der Wee, Alrijne Hospital Location Leiderdorp

Comments to the Author:

I appreciate the thorough revision the authors have performed, taking into account all the reviewers' comments. The review has clearly improved, especially the discussion section with regards to limitations and future policy. I would still advise a few minor adjustments.

Methods

p.4, line 20: I would suggest changing "in order ... in SSA" to because we wanted to provide a recent overview and explain that this makes studies before 2000 irrelevant.

We have replaced with: 'We limited the search to references published after January 1st, 2000 in order to provide a recent overview and therefore studies published before 2000 were not relevant for this purpose.'

Results

p. 11, line 42 (Characteristics of district hospitals): Elsewhere you have used the term "district hospitals" instead of the acronym DH, however, in this section you suddenly use DH again.

We have replaced 'DH' with 'district hospital'.

Discussion:

I would suggest some adjustments to the structure of the discussion:

For a clearer structure, move the strengths and limitations paragraph to the end of the discussion, as the last paragraph.

We have moved the strengths and limitations paragraph to the end of the Discussion section.

p. 17, line 57: This sentence ("this is the first review... to our knowledge) should be at the end of the limitations section, starting with "However...", explaining the importance of the study despite its limitations.

We have moved the sentence so that the limitations paragraph ends with: 'However, this is the first review to systematically assess the literature for the scope, volume and quality of surgery in district hospitals in sub-Saharan Africa, to our knowledge.'

Reviewer: 3

Competing interests of Reviewer: None declared

Thank you to the Editors and Reviewers for your helpful comments and opportunity to improve the paper.